The role of lncRNAs related ceRNA regulatory network in multiple hippocampal pathological processes during the development of perioperative neurocognitive disorders

Zhou Bowen 1
Zheng Yuxiang 1
Suo Zizheng 1
Zhang Mingzhu 1
Xu Wenjie 1
Wang Lijuan 1
Ge Dazhuang 1
http://orcid.org/0000-0001-5919-8283 Qu Yinyin 2 quyinyin@bjmu.edu.cn
Wang Qiang 1
Zheng Hui 1
Ni Cheng 1 nicheng@cicams.ac.cn
1 Department of Anesthesiology, National Cancer Center/National Clinical Research Center for Cancer/Cancer Hospital, Chinese Academy of Medical Sciences and Peking Union Medical College , Beijing , China
2 Department of Anesthesiology, Peking University Third Hospital , Beijing , China
Lasseigne Brittany
Electronic publication date: 2024 Aug 9
Publication date: 2024
Volume: 12
Electronic Location ID: e17775
Received 2023 Dec 21; Accepted 2024 Jun 28
Copyright: © 2024 Zhou et al.
Copyright year: 2024
Copyright holder: Zhou et al.
License: This is an open access article distributed under the terms of the Creative Commons Attribution License, which permits unrestricted use, distribution, reproduction and adaptation in any medium and for any purpose provided that it is properly attributed. For attribution, the original author(s), title, publication source (PeerJ) and either DOI or URL of the article must be cited.
License URL: https://creativecommons.org/licenses/by/4.0/

Keywords: Perioperative neurocognitive disorders, LncRNA, CeRNA network, Neurological system alternation, Metabolism alternation

Funding: National Natural Science Foundation of China 82171195, 82201337 and 81771146 Non-profit Central Research Institute Fund of Chinese Academy of Medical Sciences 2023-JKCS-25 Beijing Natural Science Foundation 7232131 Beijing Hope Run Special Fund of Cancer Foundation of China LC2020A01 Talent Project of National Cancer Center/Cancer Hospital Chinese Academy of Medical Sciences This work was supported by the National Natural Science Foundation of China (Nos. 82171195, 82201337 and 81771146), the Non-profit Central Research Institute Fund of Chinese Academy of Medical Sciences (No. 2023-JKCS-25), Beijing Natural Science Foundation (No. 7232131), Beijing Hope Run Special Fund of Cancer Foundation of China (No. LC2020A01), and the Talent Project of National Cancer Center/Cancer Hospital Chinese Academy of Medical Sciences (For Dr. Cheng Ni). The funders had no role in study design, data collection and analysis, decision to publish, or preparation of the manuscript.

==============================
Background

Perioperative neurocognitive disorders (PND) refer to neurocognitive abnormalities during perioperative period, which are a great challenge for elderly patients and associated with increased morbidity and mortality. Our studies showed that long non-coding RNAs (lncRNAs) regulate mitochondrial function and aging-related pathologies in the aged hippocampus after anesthesia, and lncRNAs are associated with multiple neurodegenerations. However, the regulatory role of lncRNAs in PND-related pathological processes remains unclear.

Methods

A total of 18-month mice were assigned to control and surgery (PND) groups, mice in PND group received sevoflurane anesthesia and laparotomy. Cognitive function was assessed with fear conditioning test. Hippocampal RNAs were isolated for sequencing, lncRNA and microRNA libraries were constructed, mRNAs were identified, Gene Ontology (GO) analysis were performed, and lncRNA-microRNA-mRNA networks were established. qPCR was performed for gene expression verification.

Results

A total of 312 differentially expressed (DE) lncRNAs, 340 DE-Transcripts of Uncertain Coding Potential (TUCPs), and 2,003 DEmRNAs were identified in the hippocampus between groups. The lncRNA-microRNA-mRNA competing endogenous RNA (ceRNA) network was constructed with 29 DElncRNAs, 90 microRNAs, 493 DEmRNAs, 148 lncRNA-microRNA interaction pairs, 794 microRNA-mRNA interaction pairs, and 110 lncRNA-mRNA co-expression pairs. 795 GO terms were obtained. Based on the frequencies of involved pathological processes, BP terms were divided into eight categories: neurological system alternation, neuronal development, metabolism alternation, immunity and neuroinflammation, apoptosis and autophagy, cellular communication, molecular modification, and behavior changes. LncRNA-microRNA-mRNA ceRNA networks in these pathological categories were constructed, and involved pathways and targeted genes were revealed. The top relevant lncRNAs in these ceRNA networks included RP23-65G6.4, RP24-396L14.1, RP23-251I16.2, XLOC_113622, RP24-496E14.1, etc., and the top relevant mRNAs in these ceRNA networks included Dlg4 (synaptic function), Avp (lipophagy), Islr2 (synaptic function), Hcrt (regulation of awake behavior), Tnc (neurotransmitter uptake).

Conclusion

In summary, we have constructed the lncRNA-associated ceRNA network during PND development in mice, explored the role of lncRNAs in multiple pathological processes in the mouse hippocampus, and provided insights into the potential mechanisms and therapeutic gene targets for PND.

Introduction

Long non-coding RNAs (lncRNAs) are a group of transcripts that are at least 200 nucleotides long without the capacity for protein coding due to significantly disrupted open reading frames. LncRNAs play crucial roles in regulating genomic expression, targeting and modifying chromatin complexes to silence, enhancing the expression of co-located genes, and binding to specific proteins to act as decoys, scaffolds, or guides (Wang & Chang, 2011). One of the most well-known regulatory roles of lncRNAs is through competing endogenous RNA (ceRNA) activity, whereby lncRNAs vie with mRNA transcripts for microRNA (miRNA) engagement, influencing the interaction of RNAs and subsequent regulation of gene expression (Salmena et al., 2011). The regulation of neuronal function is heavily influenced by the expression of numerous genes involved in various neuropathological processes, which control neural circuitry (Ramos et al., 2015; Roberts, Morris & Wood, 2014). Studies indicated the significant roles of lncRNAs in modulating cellular and molecular activities in brain, establishing links to several neural diseases, including Alzheimer’s disease (AD) (Wu et al., 2013), autism spectrum disorder (ASD) (Tang, Yu & Yang, 2017), schizophrenia (SZ) (Barry et al., 2014), and cognitive disorders (Feng et al., 2021). The lncRNA BC200, for instance, is considered the most critical lncRNA in the pathogenesis of AD, as it contributes to amyloid plaque formation and subsequent AD (Mus, Hof & Tiedge, 2007; Tiedge, Chen & Brosius, 1993). Furthermore, in patients with ASD, differentially expressed (DE) lncRNAs have been identified, which are enriched at protein-coding gene loci associated with brain development (Ziats & Rennert, 2013).

The demographic shift towards an older population has precipitated a notable increase in the frequency of elderly individuals undergoing a variety of surgical procedures annually, with a particular rise in the proportion of major surgery (Borchers et al., 2021). Surgical procedures could lead to a range of central nervous system (CNS) lesions, such as oxidative stress (Rosenfeldt et al., 2013), blood-brain barrier (BBB) damage (Ni et al., 2019), and neuroinflammation (Ma et al., 2017). Anesthetic agents, particularly inhaled anesthetics, are associated with mitochondrial and calcium homeostasis disorders, and synaptic dysfunction (Zhang et al., 2022b). The confluence of surgical trauma and anesthetics exposure could intensify the abnormal expression and phenotype of tau protein and mitochondrial dysfunction, leading to neurological complications characterized by symptoms such as conscious state change, cognitive disorders, and neurological dysfunction (Li et al., 2014a). This complication, known as perioperative neurocognitive disorder (PND) includes postoperative delirium (POD) and postoperative cognitive dysfunction (POCD), was defined in 2018 (Evered et al., 2018). The prevalence of PND is increasing significantly, affecting approximately 29% of patients (Borchers et al., 2021). The incidence in patients undergoing cardiac surgery was higher than others, and the longest duration was up to 7.5 years postoperatively (Evered et al., 2016). Factors such as aging, surgical procedures and anesthesia have been identified as risk contributors to PND (Kotekar, Shenkar & Nagaraj, 2018). Since PND could impede postoperative recovery and elevate mortality rates, extensive research is essential to elucidate its pathogenesis. The expression pattern of lncRNAs is related to acute CNS lesions and processes including neuronal inflammation, which is the basis of the PND process (Vivinetto et al., 2020).

Considering the regulatory roles of lncRNAs in various CNS lesions (Xiao et al., 2023; Yang et al., 2022), investigation of their roles in PND pathogenesis is crucial. According to the ceRNA hypothesis proposed by Salmena et al. (2011), ceRNA can orchestrate the transcriptomic landscape via interactions with target microRNAs through microRNA response elements. Recent studies have suggested that lncRNAs, such as OIP5-AS1 (Sun et al., 2022), GAS5 (Zhang et al., 2022a), and NONMMUT055714 (Wei et al., 2021), played important roles in PND. However, lncRNA-associated ceRNA networks for the pathological processes in the hippocampus during PND are still lacking. As the regulatory roles of lncRNAs are marked by their complexity, encompassing upstream and networked interactions, this study was designed to explore their effects on PND from a global perspective. We established lncRNA-associated ceRNA networks in PND pathogenesis, focusing on neuroinflammation, metabolism and neuronal development. The results provide a theoretical basis and lncRNA-associated therapeutic targets for PND.

Materials and Methods

Surgery and anesthesia

The animal experiments were performed following the guidelines for the care and use of laboratory animals and the protocol was approved by the local biomedical ethics committee (NCC2021A040). Female C57BL/6 mice, 18 months, weighing between 23 and 34 g were used. The mice were housed in cages and maintained in a standard housing condition with food and water ad libitum for 2 weeks. All mice were randomly assigned to control group and surgery (PND) group. Since PND pathogenesis occurred within postoperative day 1, and PND symptoms gradually developed during postoperative day 1 to 7 (Jin, Hu & Ma, 2020), we chose postoperative day 1 as the research node for RNA sequencing (RNA-seq) and lncRNAs related ceRNA network construction, and postoperative day 2 and 7 for cognitive function evaluation.

The mice in PND group received 2.5% sevoflurane in 50% oxygen (The MAC of sevoflurane for mice was 2.4–2.7% (Li et al., 2014c)), and the concentration of sevoflurane was monitored with an anesthetic monitor (Datex, Tewksbury, MA, USA). The exploratory laparotomy was performed as described in previous studies (Han et al., 2020; Zhang et al., 2023), in which a longitudinal midline incision was made from xiphoid to 0.5 cm proximal pubic symphysis on the skin. Each layer of abdominal muscles and peritoneum was exteriorized, and about 10 cm of the intestine was exposed to air extracorporeally. The bowel loops remained outside the abdominal cavity for 1 min and then replaced into the abdominal cavity. Finally, the incision was sutured layer by layer with 5–0 Vicryl thread. The rectal temperature was maintained at 37 ± 0.5 °C. Then the mice were put into a chamber containing 50% oxygen until 10 min after the recovery of consciousness. This surgery did not affect blood pressure and blood gas in preliminary experiments. The mice in control group did not receive anesthesia or surgical procedures, and the rest of interventions were the same in both groups.

Fear conditioning test (FCT)

The FCT (Xeye CPP; MacroAmbition S&T Development, Beijing, China) was used to assess the cognitive function of mice after surgery as described in our previous studies (Suo et al., 2022). FCT consisted of a training process 3 h after surgery and evaluations 2 and 7 days after surgery. In the training process, mice were placed in the context chamber to acclimate for 180 s, then they received a 2 Hz pulsating tone (80 dB, 3,600 Hz) for 60 s co-terminated with a mild foot shock (0.8 mA, lasting 0.5 s). In the evaluations, the hippocampus-dependent memory was assessed by the freezing time during exposure to a novel context test (the test was performed in the same chamber but with no cues or shock), while the hippocampus-independent memory was assessed by the freezing time during exposure to the tone stimulus (the test was performed in an alternative context and with no shock).

RNA extraction

The mice were sacrificed by decapitation 24 h after surgery. The brain tissues were removed and hippocampi were dissected out. Total RNAs were isolated from hippocampi with TRIzol reagent (Invitrogen, Carlsbad, CA, USA), and digested with RNase-Free DNase to remove residual DNAs. The Quantity and purity were detected with Nanodrop 2100 (Thermo Fisher, Wilmington, DE, USA) and Qubit Fluorometer (Invitrogen, Carlsbad, CA, USA). Sequencing libraries were generated with NEBNext® UltraTM RNA Library Prep Kit for Illumina® (NEB, USA) following the manufacturer’s recommendations, and index codes were added to attribute sequences to each sample.

LncRNA library construction

Total RNAs were enriched by magnetic beads with Oligo (dT) and interrupted into short segments with the addition of a fragmentation buffer after samples were qualified. Subsequently, double-stranded cDNA was synthesized with M-MuLV Reverse Transcriptase (RNase H-) and DNA Polymerase I and RNaseH, subjected to terminal reparation, and serial sequencing. The library fragments were purified with the AMPure XP system (Beckman Coulter, Brea, CA, USA) to select cDNA fragments of preferentially 150–200 bp in length. Selected double-stranded cDNA was subjected to PCR enrichment, construction, and sequencing of the RNA-seq library for each sample was conducted (Compass Biotechnology, Beijing, China) based on the protocols of Illumina HiSeqTM2500/MiSeq™ to generate paired-end reads (125/150 bp in length). The quality of RNA-seq reads from all the brain tissues was checked using FastQC (v0.11.5; Babraham Institute, Cambridge, UK). At the same time, Q20, Q30, and GC content of the clean data were calculated, and all the downstream analyses were based on clean data.

All transcriptions were merged by Cuttmerge (v2.0.2), and those with uncertain chain directions were removed. All the candidates with exons ≥2 and lengths ≥200 bp were selected for subsequent analysis. With the Cuffcompare (v2.1.1) software, transcripts that overlapped with the exon region of the database were eliminated of lncRNAs that may have coding potential. Since the abundance of transcription can directly indicate the level of expression, and FPKM considers the effect of sequencing depth and gene length for the reads count at the same time, which is currently the most commonly used method for estimating gene expression levels, FPKM was selected to represent the expression level of genes. FPKM ≥ 0.5 of all the selected transcripts were chosen with Cuffquant to calculate the expression of each transcript, and coding potentials were forecast by CNCI (v2), CPC2 (cpc-0.9-r2) and PhyloCSF (20121028). All the transcripts without coding potential were included as lncRNAs in the succeeding analysis, and those transcripts forecast by only one software were brought into analysis and marked as the TUCP.

All RNA data was analyzed by known annotated lncRNA databases (starbase v2.0) (Li et al., 2014b). Reference genome and gene model annotation files were downloaded directly from the genome website. The index of the reference genome was built using STAR and paired-end clean reads were aligned with the reference genome using STAR (v2.5.1b). STAR used the method of Maximal Mappable Prefix (MMP), which generates precise mapping results for junction reads.

MicroRNA library construction

Since the microRNA has the characteristic structure of the 3′ and 5′ ends (complete phosphate groups at the 5′ end and hydroxyl groups at the 3′ end), total RNAs were added with connectors by using Small RNA Sample Pre Kit (Illumina, San Diego, CA, USA). Then, total RNAs were inverting transcription to synthesize cDNA. Subsequently, cDNA was amplified by PCR, all interested cDNA was isolated using PAGE-glue electrophoresis. According to the manufacturer’s instructions, the libraries were sequenced on an Illumina HiSeq 2500 platform (Illumina, San Diego, CA, USA). The quality of RNA-seq reads from all hippocampus was checked using FastQC (v0.11.5, Babraham Institute, Cambridge, UK).

DEmRNAs identification and GO/KEGG analysis

Differential expression analysis of two groups was performed using the edgeR package (3.12.1, for lncRNAs and mRNAs) or DESeq2 R package (1.10.1, for microRNAs). The codes used in the study can be accessed in the Supplemental Files. All the p-values were adjusted using the Benjamini & Hochberg method. The results were standardized with FPKM (lncRNAs and mRNAs) or TMM (microRNAs), the significance and fold-change were set as p < 0.05 and log2 |fold change| > 1.

The overall distribution of the differentially expressed genes (DEGs) was shown by volcano plots. Gene Ontology (GO) and KEGG functional annotation enrichment analyses were performed for DEmRNAs using the ClusterProfiler R package (v4.0.1). GO enrichment analysis contains three categories: biological process, molecular function, and cellular component. The enriched GO terms of biological processes were analyzed and grouped into eight categories. The top 10 terms with p < 0.05 and the numbers of genes were explored. The results were shown with bubble charts, and the significance was given priority.

CeRNA network construction

The lncRNA-microRNA-mRNA ceRNA network was based on the theory that lncRNAs regulate mRNA activity through invoking microRNA sponges (Li et al., 2014b). All DElncRNAs and DEmRNAs were analyzed for ceRNA network construction. Pearson correlation coefficient was used to analyze lncRNA-mRNA co-expression, and the significance was set as p < 0.05. Pearson correlations were calculated using log10 (FPKM-UQ). The interaction between microRNA and mRNA was predicted by binding energy with three software based on sequencing results (miRanda (Betel et al., 2010), PITA (Kertesz et al., 2007), and RNAhybrid (Kruger & Rehmsmeier, 2006)). A microRNA-mRNA pair was considered if all three software prediction results were significant. The lncRNA-microRNA interaction was detected by miRANDA since the binding energy was calculated using sequencing results.

DE-RNAs were further applied to construct eight ceRNA networks and reveal lncRNA-associated regulation in the eight categories of biological processes. The DEmRNAs were used to search relevant lncRNAs and microRNAs by Perl package (v5.32.1.1). LncRNA-mRNA pairs, lncRNA-microRNA pairs, microRNA-mRNA pairs, and lncRNA-microRNA-mRNA pairs were shown in each ceRNA network diagram with Cytoscape (3.8.2).

Quantitative real-time PCR (qPCR)

qPCR was performed on CFX96 Real-Time PCR Detection System (Bio-Rad, Hercules, CA, USA). Amplification mixture consisted of PowerUpTM SYBRr Green master mix (Thermo Fisher, Waltham, MA, USA), 10 µM forward and reverse primers (Invitrogen, Carlsbad, CA, USA), and approximately 1.5 µl of cDNA template. Primer sequences were obtained from the literature and checked for their specificity through in silico PCR. The forward and reverse primers are shown in Table 1. Amplification was carried out with an initial denaturation step at 95 °C for 2 min, followed by 45 cycles of 95 °C for 10 s, 55 °C for 30 s and 60 °C for 30 s, then 65 °C for 2 min in 10 µl reaction volume. All reactions were run in duplicate and the results were averaged from 6 independent studies. qPCR was quantified in two steps. First, β-actin levels were used to normalize target gene levels (ΔCt = Cttarget gene − Ctβ-actin, target gene level = 2−ΔCt). Second, the target gene levels of PND group were presented as the percentage of control group, and 100% of the target gene levels referred to control levels.

Table 1 The forward and reverse primers for qPCR.

Genes	Primers	Sequence (5′ to 3′)	
Dlg4	Forward primer	GGCTTCATTCCCAGCAAACG	
	Reverse primer	CATCGTTGGCACGGTCTTTG	
Hcrt	Forward primer	CTCCTTCAGGCCAACGGTAA	
	Reverse primer	GGGTGCTAAAGCGGTGGTA	
Robo2	Forward primer	TTGCGAATTGTTCATGGGCG	
	Reverse primer	ATGGTTGGTTCTGGGTGTCC	
Islr2	Forward primer	TGCAGACTGTGCCTACAAGG	
	Reverse primer	CTGACTCTACCGTGCGTACC	
Yam1	Forward primer	TCAATCTCGGGTGGCTGAAC	
	Reverse primer	TCAACACGGGAAACCTCACC	
Lhx1os	Forward primer	GCCTCCCAGAGAAGTGTGAA	
	Reverse primer	TACACACTGCAGACAGCTACA	
Malat1	Forward primer	GATTTCTCTGCCACATCGCC	
	Reverse primer	TGGGCATAACCTTGAAACCGA	
Lsmem2	Forward primer	TGTCTTACCACCATCGCTGT	
	Reverse primer	CTAGCCAGACGCAATTTGAGC	

Statistical analysis

To ensure the accuracy of the research results, we applied appropriate methods to adjust the p-value, thereby reducing the false discovery rate (FDR). The correction method has been described in the text. The statistical calculations were performed with GraphPad Prism 7.0 software (for FCT and qPCR results) or R (for gene enrichment analysis). Quantitative data were presented as mean ± standard deviation (SD). Non-paired double-tailed Student’s t-test was used to identify significant differences between two groups. A p-value < 0.05 was considered statistically significant. The significance of GO and KEGG enrichment analysis was calculated by the hypergeometric distribution and Fisher exact test, and lower p-value indicated that the term was more significantly enriched.

Results

Cognitive function assessed by fear condition test in aged mice

The aged mice were subjected to control and PND groups (n = 12), and the FCT was used to assess the cognitive function. In the context test, the freezing time decreased significantly at 2 days (48.10 ± 17.92 vs. 28.88 ± 11.19, p = 0.0046, Fig. 1A) and 7 days after surgery (36.33 ± 13.01 vs. 22.41 ± 9.33, p = 0.0064, Fig. 1B) in PND group. In the tone test, there was no significant between two groups at 2 days (62.27 ± 24.80 vs. 47.29 ± 18.85, p = 0.1097, Fig. 1C) or 7 days after surgery (44.37 ± 17.29 vs. 33.77 ± 14.51, p = 0.1181, Fig. 1D). These results indicated the occurrence of hippocampus-dependent cognitive dysfunction in PND group.

Figure 1 Fear conditioning test results.

Fear conditioning test: the freezing time of context test in postoperative day 2 (A) and 7 (B), and tone test in postoperative day 2 (C) and 7 (D). **, p < 0.01.

DElncRNAs, DE-TUCPs, and DEmRNAs in the hippocampus

The hippocampus of aged mice in control and PND groups were collected, and RNA-seq was used to analyze the expression of lncRNAs, microRNAs, and mRNA (coding genes) in the hippocampus. Volcano plots illustrating the differential expression of lncRNAs (Fig. 2A), Transcripts of Uncertain Coding Potential (TUCPs) (Fig. 2B), and mRNAs (Fig. 2D), with blue dots indicating down-regulation (p < 0.05) and red dots indicating up-regulation (p < 0.05). Compared with control group, there were 312 DElncRNAs (P < 0.05), with 154 down-regulated and 158 up-regulated in PND group (Fig. 2A). TUCPs are another set of lncRNAs that have been shown to have the potential to encode peptides (Kondo et al., 2010), but are excluded by Pfam scan criteria during filtering steps (Cabili et al., 2011). TUCPs play roles in gene expression regulation as part of pseudogenes (Salih et al., 2019). Compared to other lncRNAs, TUCPs are more conserved, suggesting that they are a complementary part of the regulation of ceRNA network, and could play roles in PND pathogenesis. There were 340 differentially expressed TUCPs (DE-TUCPs) (P < 0.05) in the PND group compared to the control, with 141 down-regulated and 199 up-regulated (Fig. 2B). Since TUCPs may contain a short open reading frame, their regulatory role could be achieved as a lncRNA or a small peptide. Hence, this study excluded TUCPs from subsequent ceRNA network analysis to focus on their distinct functions outside the network. DElncRNAs were classified based on the positional relationship of the transcripts and their corresponding genes. There were 197 intronic lncRNAs (65%), 81 intergenic lncRNAs (lincRNAs, 27%), 24 antisense lncRNAs (8%), and 10 lncRNAs with multiple positions and uncertain classification (3%, Fig. 2C). Compared to control group, there were 2,003 DEmRNAs of protein-coding genes (P < 0.05) with 1,180 down-regulated and 823 up-regulated in the PND group (Fig. 2D).

Figure 2 DElncRNAs, DETUCPs and DEmRNAs in PND group.

The DElncRNAs (A), DETUCPs (B), the classification of DElncRNAs (C) and DEmRNAs (D) in PND group. The red dots are up-regulated genes and the blue ones are down-regulated genes.

Interaction among DElncRNA, DEmiRNA and DEmRNA

In ceRNA regulatory network, lncRNAs competitively bind to microRNAs and regulate downstream transcription and translation. This study focused on the potential regulatory roles of lncRNAs in ceRNA networks of different physiological processes. All miRNAs were analyzed for interaction pairs and for constructing the ceRNA network, regardless of whether there is any significance. The binding energy between lncRNAs and microRNAs was calculated using miRANDA software with a binding energy threshold of less than −20 kcal/mol considered significant. Consequently, 29 DElncRNAs, 90 microRNAs and 148 lncRNA-microRNA interaction pairs were targeted. RP24-396L14.1 and RP23-65G6.4 had the largest and second-largest number of co-expression pairs with microRNAs. The lncRNA-microRNA interaction network is shown depicted in Fig. 3. The microRNA-mRNA interactions were also determined by binding energy. The microRNA-mRNA interactions were also determined by binding energy. Similarly, miRNA-mRNA interactions were determined based on binding energy, with a threshold of less than −10 kcal/mol deemed significant. Then, 145 microRNAs, 608 DEmRNAs and 1,103 microRNA-mRNA interaction pairs were targeted. The microRNA-mRNA interaction network is illustrated in Fig. 4.

Figure 3 LncRNA-microRNA regulatory network.

The lncRNA-microRNA regulatory network during PND development included 29 DElncRNAs, 90 microRNAs, and 148 interaction pairs. The red dots represent DElncRNAs, and the yellow dots represent microRNAs. Sizes of dots represent the number of pairs related to this RNA.

Figure 4 MicroRNA-mRNA regulatory network.

The microRNA-mRNA regulatory network during PND development included 145 microRNAs, 608 DEmRNAs, and 1,102 interaction pairs. The yellow dots represent microRNAs, and the blue dots represent DEmRNAs. Sizes of dots represent the number of pairs related to this RNA.

LncRNAs could also regulate mRNA expression through other mechanisms, including acting as decoys, scaffolds or guides (Pant et al., 2021). Thus, we further analyzed the relationships between DElncRNAs and DEmRNAs. Significant correlations were identified based on PEARSON correlation with coefficients >0.95 and p-value < 0.05. In total, 20 annotated DElncRNAs, 219 novel DElncRNAs and 1,200 DEmRNAs were involved 14,053 lncRNA-mRNA co-expression pairs were targeted, including 2,859 annotated lncRNA-mRNA pairs and 11,194 novel lncRNA-mRNA pairs. The top annotated and novel lncRNAs (ranked by the number of co-expression pairs) are illustrated in Fig. 5, with 216 annotated lncRNA-mRNA pairs and 955 novel lncRNA-mRNA pairs. Among these, RP23-65G6.4 and XLOC_113622 had the largest number of co-expression pairs with mRNAs among annotated lncRNAs and novel lncRNAs, respectively, and both of them are closely related to behavior alteration and neuronal development. Due to the vast number of lncRNA-mRNA co-expression pairs, lncRNAs play roles across various neural functions in PND pathogenesis, highlighting the complexity of the lncRNA-associated regulatory network.

Figure 5 LncRNA-mRNA regulatory network.

The lncRNA-mRNA regulatory network during PND development included 20 annotated DElncRNAs and 219 novel DElncRNAs, 1,200 DEmRNAs, and 1,195 pairs of interaction. The red dots represent DElncRNAs, and the blue dots represent DEmRNAs. Sizes of dots represent the number of pairs related to this RNA.

LncRNA-microRNA-mRNA ceRNA network in the hippocampus

Considering that the ceRNA network is the most fundamental way of lncRNA regulation, based on lncRNA-microRNA interaction pairs and microRNA-mRNA interaction pairs, the lncRNA-microRNA-mRNA ceRNA network in the hippocampus during PND development was constructed (Fig. 6). The ceRNA network included 29 DElncRNAs, 90 microRNAs, 493 DEmRNAs, 148 lncRNA-microRNA interaction pairs, and 794 microRNA-mRNA interaction pairs. Additionally, through lncRNA-mRNA co-expression analysis within this network, 110 lncRNA-mRNA co-expression pairs were identified. The top 10 DElncRNAs involved were RP24-396L14.1, RP23-134H19.3, RP23-250A14.2, RP23-65G6.4, AC120859.3, RP23-366O14.4, RP23-251I16.2, RP24-252B21.2, Yam1, Malat1. Moreover, the top 10 DEmRNAs were Vgf, Apol9b, Dpp6, Myo18a, Osbpl5, Rps6ka1, Psap, Cckar, Ccnd3, Abcc10.

Figure 6 LncRNA-microRNA-mRNA ceRNA network.

This lncRNA-microRNA-mRNA ceRNA network during PND development included 29 DElncRNAs, 90 microRNA, 493 DEmRNAs, 148 lncRNA-microRNA interaction pairs, and 794 microRNA-mRNA interaction pairs. The red dots represent DElncRNAs, the yellow dots represent DEmicroRNAs, and the blue dots represent DEmRNAs. Sizes of dots represent the number of pairs related to this RNA.

With GO enrichment analysis, we found that lncRNAs could regulate various biological processes through ceRNA networks, primarily in neurological system alteration, neuronal development and behavior alteration. These results indicated possible roles and involved ceRNA networks of lncRNAs during PND development.

Functional enrichment analysis and subclassification

GO analysis indicated 795 enriched GO terms of DEmRNAs, including 619 biological process (BP), 68 cellular component (CC) and 108 molecular function (MF) terms. The top 10 BP, CC and MF terms are presented as bubble charts in Figs. 7A–7C. The top three enriched BP terms were positive regulation of cellular process, positive regulation of biological process, and regulation of cellular component organization. The enriched CC terms included multiple intracellular cellular organelles and membrane-bounded organelles. The enriched MF terms included multiple intracellular molecular modification processes, and the top three terms were neurohypophyseal hormone activity, receptor binding, and neuropeptide hormone activity. Based on the frequencies of involved pathological processes, BP terms were further classified into eight categories: neurological system alteration, neuronal development, metabolism alteration, immunity and neuroinflammation, apoptosis and autophagy, cellular communication, molecular modification, and behavior changes. The top 10 BP terms within each category were presented as bubble charts in Figs. 7D–7K.

Figure 7 GO and KEGG analysis of DEmRNAs.

Bubble charts show GO enrichment analysis of DEmRNAs in biological process (A), cellular component (B), molecular function (C), and categories of neurological system alternation (D), neuronal development (E), metabolism alternation (F), immunity and neuroinflammation (G), apoptosis and autophagy (H), cellular communication (I), molecular modification (J), and behavior changes (K), KEGG analysis result was also demonstrated by bubble chart (L). The horizontal axis means the gene ratio, equal to sample frequency/background frequency. Color in blue means low p-value and the area of bubbles means the gene count.

For the category neurological system alteration, there were 25 enriched GO terms. The top three GO terms were regulation of neurological system process, glial cell proliferation, and axon extension (Fig. 7D). The lncRNA-microRNA-mRNA ceRNA network in neurological system alteration was constructed (Fig. 8A), which included 110 DElncRNAs (18 annotated lncRNAs and 92 novel lncRNAs), 15 microRNAs, 25 DEmRNAs, 316 lncRNA-mRNA co-expression pairs, 23 microRNA-mRNA interaction pairs, 16 lncRNA-microRNA interaction pairs, and five lncRNA-microRNA-microRNA ceRNA regulatory pathways. Notably, all five pathways interacted with Dlg4, which encoded postsynaptic density-95 and regulated important molecules for neuronal function (Bustos et al., 2017; Zhang et al., 2017, 2023). RP23-65G6.4 emerged as a pivotal lncRNA within this network, regulating the expressions of Dlg4 and other mRNAs via microRNAs such as mmu-miR-361-3p and mmu-miR-665-3p, highlighting its potential role in PND-related neurological system alteration. For neuronal development, 26 enriched GO terms were identified, which mainly cover the process of neuronal genesis and differentiation, especially the process of synaptogenesis. The top three GO terms were cell morphogenesis involved in axon development, nervous system development, and neuron differentiation (Fig. 7E). The corresponding ceRNA network for neuronal development (Fig. 8B) included 116 DElncRNAs (12 annotated lncRNAs and 104 novel lncRNAs), 28 microRNAs, 29 DEmRNAs, 293 lncRNA-mRNA pairs, 35 microRNA-mRNA pairs, and 15 lncRNA-microRNA pairs. The most relevant mRNAs in neuronal development were Tnc and Slc6a3, which participated in nervous system axon regeneration and neurotransmitter uptake (Chen et al., 2010; Reith et al., 2022). These mRNAs were regulated by RP23-65G6.4 and RP24-396L14.1, which were the top two relevant lncRNAs, underpinning their significance in PND-related neuronal development. The regulatory role of lncRNA-related ceRNA networks in neuronal development has also been confirmed in previous studies (Plasil et al., 2022).

Figure 8 LncRNA-microRNA-mRNA ceRNA networks in category neurological system alternation, etc.

The lncRNA-microRNA-mRNA ceRNA networks in category neurological system alternation (A), neuronal development (B), metabolism alternation (C), and immunity and neuroinflammation (D), during PND development. The red dots represent DElncRNAs, yellow dots represent DEmicroRNAs, and the blue dots represent DEmRNAs. Sizes of dots represent the number of pairs related to this RNA.

In the category of metabolism alteration, there were 75 enriched GO terms. The top three GO terms were regulation of metabolic process, regulation of macromolecule metabolic process, and regulation of primary metabolic process (Fig. 7F). The ceRNA network related to metabolism alteration was identified as the most enriched ceRNA network among eight categories (Fig. 8C), which included 159 DElncRNAs (145 annotated lncRNAs and 14 novel lncRNAs), 64 microRNAs, 76 DEmRNAs, 684 lncRNA-mRNA pairs, 119 microRNA-mRNA pairs, 53 lncRNA-microRNA pairs, and 10 lncRNA-microRNA-microRNA ceRNA network regulatory pathways. Notably, lncRNAs RP23-65G6.4 and RP24-396L14.1 were implicated in the regulation of Ank1, which is crucial to AD pathogenesis, and hypermethylated Ank1 gene domain was found in the entorhinal cortex of AD patients (De Jager et al., 2014). For neuroimmunity and inflammation, there were 21 enriched GO terms covering glial cell proliferation and functional regulation. The top three GO terms were microglial cell proliferation, macrophage proliferation and microglial cell activation (Fig. 7G). The corresponding ceRNA network in immunity and neuroinflammation (Fig. 8D) included 48 lncRNAs (Five annotated lncRNAs and 43 novel lncRNAs), seven microRNAs, eight mRNAs, 51 lncRNA-mRNA pairs, and seven microRNA-mRNA pairs. The most relevant mRNA was Ppp3cb, which enables calmodulin-dependent protein phosphatase activity and axon guidance receptor activity (Pun et al., 2022). All the tops three relevant lncRNAs in this network interacted with Ppp3cb, Including XLOC_085831, XLOC_113622, and RP24-351J24.2. Multiple novel lncRNAs were also relevant with specific mRNAs to produce corresponding regulatory effects in this network.

For apoptosis and autophagy, there were 13 enriched GO terms. The top three GO terms were negative regulation of intrinsic apoptotic signaling pathway in response to DNA damage, regulation of intrinsic apoptotic signaling pathway, and positive regulation of autophagy (Fig. 7H). The ceRNA network related to apoptosis and autophagy (Fig. 9A) included 78 DElncRNAs (11 annotated lncRNAs and 67 novel lncRNAs), 14 microRNAs, 11 DEmRNAs, 106 lncRNA-mRNA pairs, 14 microRNA-mRNA pairs, and 12 lncRNA-microRNA pairs. The top three relevant mRNAs were Avp, Wwox, and Rab7, which participated in the regulatory process of apoptotic process and lipophagy (Chang & Chang, 2015; Roy et al., 2013; Wahlstrom et al., 2004). This network suggests a more specific interaction landscape among lncRNA-mRNA pairs, highlighting the potential for targeted therapeutic interventions. For cell communication, there were 23 enriched GO terms with significance. The top three GO terms were regulation of cell communication, neuron projection morphogenesis, and regulation of signaling (Fig. 7I). The ceRNA network for cell communication (Fig. 9B) included 146 DElncRNAs (28 annotated lncRNAs and 132 novel lncRNAs), 55 microRNAs, 63 DEmRNAs, 635 lncRNA-mRNA pairs, 89 microRNA-mRNA pairs, 47 lncRNA-microRNA pairs, and four lncRNA-microRNA-mRNA ceRNA pathways. All of these four ceRNA pathways interacted with Ank1, Rab7 and Dlg4. The top three DEmRNAs were Dlg4, Peg10 and Ank1. Peg10 was involved in vesicle-mediated intercellular transport and was widely expressed in the nervous system (Herrera-Barrera & Sahay, 2022; Zaitoun et al., 2010). The lncRNAs interacted with Peg10 including RP24-396L14.1, XLOC_113622 and Lhx1os, while the former two lncRNA were also in the top three relevant lncRNAs list of this ceRNA regulatory network.

Figure 9 LncRNA-microRNA-mRNA ceRNA networks in category apoptosis and autophagy, etc.

The lncRNA-microRNA-mRNA ceRNA networks in category apoptosis and autophagy (A), cellular communication (B), molecular modification (C), behavior changes (D) during PND development. The red dots represent DElncRNAs, yellow dots represent DEmicroRNAs, and the blue dots represent DEmRNAs. Sizes of dots represent the number of pairs related to this RNA.

For modification pathways, there were 18 enriched GO terms. The top three GO terms were protein K6-linked ubiquitination, mRNA modification, and regulation of post-translational protein modification (Fig. 7J). The ceRNA network for molecular modification (Fig. 9C) included 111 DElncRNAs (11 annotated lncRNAs and 100 novel lncRNAs), 17 microRNAs, 15 DEmRNAs, 164 lncRNA-mRNA pairs, 20 microRNA-mRNA pairs, and 13 lncRNA-microRNA pairs. Despite the broad involvement of 111 DElncRNAs, interactions predominantly focused around a few mRNAs, the top three relevant mRNAs were Dlg4, Slc35c2 and Avp. Slc35c2 is a type of GDP-fucose transporter implicated in the Notch signaling pathway through regulating fucosylation of Notch receptors (Lu et al., 2010), thus participating in various central nervous system diseases, such as Alzheimer’s disease (Ma et al., 2022). XLOC_113622, the most relevant lncRNA in this ceRNA network, interacted with Slc35c2, highlighting its potential regulatory impact on the PND-related molecular modification. For behavioral alteration, there were 48 enriched GO terms. The top three GO terms were maternal aggressive behavior, locomotory behavior and parental behavior (Fig. 7K). The ceRNA network in behavioral alteration (Fig. 9D) included 129 DElncRNAs (three annotated lncRNAs and 116 novel lncRNAs), 33 microRNAs, 37 DEmRNAs, 428 lncRNA-mRNA pairs, 42 microRNA-mRNA pairs, 31 lncRNA-microRNA pairs, and seven lncRNA-microRNA-microRNA ceRNA network pathways. The top three relevant mRNAs were Dlg4, Col1a1 and Tnc. Col1a1 encodes procollagen α1/α2 chains of type I collagen, which are important components of the extracellular matrix, and participate in various developmental processes (Liu et al., 2021). Tnc participated in nervous system axon regeneration and neurotransmitter uptake (Chen et al., 2010). RP24-496E14.1-mmu-miR-532-5p-Hcrt was involved in the related ceRNA network, and Hcrt was associated with sleep-wakefulness, which played a role in PND pathogenesis (Jones & Hassani, 2013; Li et al., 2022).

The top three terms from the KEGG analysis were ECM-receptor interaction, glutamatergic synapse, and PI3K-Akt signaling pathway (Fig. 7L). The ECM-receptor interaction is primarily involved in cellular communication and neuronal development. It plays a role in synaptogenesis and the formation of neural networks, thereby affecting learning and cognition. The glutamatergic synapse is mainly related to synaptic plasticity and the regulation of neurotransmitter release, and linked to various neurodegenerative changes and neurological system alteration. The PI3K-Akt signaling pathway is also involved in neuronal development, apoptosis and autophagy, which could contribute to the pathogenesis of PND (Yang et al., 2022).

Validation of DElncRNAs and DEmRNAs

The expression changes of four DElncRNAs (Yam1, Lhx1os, Malat1, and Lsmem2) and four DEmRNAs (Dlg4, Hcrt, Robo2, and Islr2) involved in these ceRNA networks were validated with qPCR. The results indicated that the expression changes of lncRNA Yam1 (Control group vs. Surgery group, 100.00 ± 20.44 vs. 149.40 ± 47.50, p < 0.05), Lhx1os (100.00 ± 62.44 vs. 38.02 ± 18.60, p < 0.05), Malat1 (100.00 ± 40.55 vs. 44.12 ± 22.79, p < 0.05), and Lsmem2 (100.00 ± 33.56 vs. 50.00 ± 21.91, p < 0.05) had a good correlation with RNA-seq results. The expression changes of Dlg4 (100.00 ± 24.82 vs. 59.57 ± 25.69, p < 0.05), Hcrt (100.00 ± 31.62 vs. 59.26 ± 25.98, p < 0.05), and Robo2 (100.00 ± 40.55 vs. 165.30 ± 52.96, p < 0.05) also had a good correlation with RNA-seq results. However, the expression changes of Islr2 were not significant after surgery (100.00 ± 54.09 vs. 106.40 ± 77.17, p = 0.87) in qPCR validation. The qPCR results are displayed in Fig. 10.

Figure 10 qPCR results of DElncRNAs and DEmRNAs.

qPCR results of four DElncRNAs (A) and four DEmRNAs (B). White bars represent the control group and blue bars represent the PND group. *p < 0.05.

Discussion

In this study, we developed PND model in aged mice and investigated the role of lncRNAs and related lncRNA-microRNA-mRNA ceRNA networks in the hippocampus. Through rigorous RNA sequencing and subsequent analyses, 312 DElncRNAs, 340 DE-TUCPs and 2,003 DEmRNAs were identified. The lncRNA-microRNA-mRNA ceRNA network was constructed with 29 DElncRNAs, 90 microRNAs, 493 DEmRNAs, 148 lncRNA-microRNA interaction pairs, 794 microRNA-mRNA interaction pairs, and 110 lncRNA-mRNA co-expression pairs. GO enrichment analysis further elucidated 795 enriched GO terms, and all BP terms were divided into eight categories based on the frequencies of involved pathological processes: neurological system alteration, neuronal development, metabolism alteration, immunity and neuroinflammation, apoptosis and autophagy, cellular communication, molecular modification, and behavior changes. LncRNA-associated ceRNA networks in these categories were constructed, then involved pathways and targeted genes were revealed. Among the findings, several lncRNAs—such as Yam1, Lhx1os, Malat1, and Lsmem2—emerged as potential therapeutic targets, underscoring the significant regulatory and therapeutic value of lncRNAs in PND pathogenesis.

LncRNAs have emerged as pivotal regulators in various neurological disorders, such as schizophrenia (Ni et al., 2021; Teng et al., 2023), AD (Wu et al., 2013), and autism spectrum disorder (Parikshak et al., 2016). Literately, DElncRNAs were classified into intronic lncRNA, lincRNA, and antisense lncRNA. The results showed that the lncRNA-mRNA co-expression pairs of intronic DElncRNAs were much specific. XLOC_001994 had an interaction pair with was noted for its interaction with Abi1, which was expressed in brain and involved in neuronal projections (Courtney et al., 2000). XLOC_013392 regulated Brd2 expression, linking it to juvenile myoclonic epilepsy (Pal et al., 2003). Almost 70% of coding genes have antisense transcripts, and antisense lncRNAs are located in these antisense transcripts (He et al., 2008). Antisense lncRNAs could increase the methylation of transcription sites, modify the post-transcriptional RNAs, and regulate the expression of the corresponding gene (Canzio et al., 2019). They could also regulate gene expression through binding microRNAs (Liu et al., 2015) or producing endogenous siRNAs (Faghihi & Wahlestedt, 2009). This study showed that all the top three relevant lncRNAs were antisense lncRNAs, suggesting the role of endogenous competitive bind and post-transcriptional modification endogenous siRNAs in PND-related gene regulation and pathogenesis. The regulatory model of antisense lncRNA has been proven to play a significant role in the pathogenesis of cognitive dysfunctions (Dos Reis et al., 2024).

LincRNA could regulate chromatin topology in trans or scaffold, the activity of proteins and RNA, and adjacent transcriptional processes (Ransohoff, Wei & Khavari, 2018). Our study showed that among the top five relevant lncRNAs, XLOC_113622 and RP24-496E14.1 were lincRNAs involved in the ceRNA networks of all eight categories. Lhx1os, identified as a lincRNA, notably participates in the ceRNA networks spanning all eight BP categories, showcasing its broad regulatory spectrum. Lhx1os interacts with microRNAs, including mmu-miR-199a-5p and mmu-miR-107-3p. Mmu-miR-199a-5p participated in the ceRNA network of category metabolism alteration, immunity and neuroinflammation, apoptosis and autophagy, cellular communication and behavior changes. At the same time, it also played roles in ischemic stroke-induced cognitive disorders (Jin et al., 2023) and posterodorsal medial amygdala regulated behavior changes (Hirsch et al., 2018). Mmu-miR-107-3p participated in the ceRNA network of category metabolism alteration and cellular communication and the metabolism process of Wilson’s disease (Wei et al., 2022). Therefore, Lhx1os could be the target for PND prediction and treatment. MALAT1 is one of the most highly expressed lincRNAs in CNS and plays a crucial role in neuronal development (Bernard et al., 2010; Chen et al., 2016; Pant et al., 2021). MALAT1 inhibited miR-125b, suppressed neuronal apoptosis and neuroinflammation, and promoted neurite outgrowth in AD (Chanda, Jana & Mukhopadhyay, 2022). In this study, we observed decreased expression of Malat1 in the hippocampus after surgery, and Malat1 was involved in the ceRNA networks of category neurological system alteration, neuronal development, and cellular communication. As lincRNA, Malat1 might act as a decoy for proteins and RNAs, positioning it as a pivotal player in PND pathology through its influence on ceRNA networks.

As described in the text, lncRNAs can regulate the expression levels of corresponding mRNAs directly or through the ceRNA mechanism (by competitively binding microRNAs), thereby influencing various biological processes. These processes include eight categories: neurological system alteration, neuronal development, metabolism alteration, immunity and neuroinflammation, apoptosis and autophagy, cellular communication, molecular modification, and behavior changes, thus widely participating in the pathogenesis of PND. In these lncRNA-associated ceRNA networks, some lncRNAs can precisely regulate the expression levels of corresponding mRNAs, while others can regulate the expression levels of multiple mRNAs, thereby participating in various biological processes with a network-like regulatory mechanism. The lncRNA-associated ceRNA network is characterized by its complexity and interconnectivity, requiring further research to elucidate deeper mechanisms.

The top five relevant mRNAs were Dlg4, Avp, Islr2, Hcrt and Tnc (ranked by the amount of RNA interaction pairs), and the mRNAs with broad involvement across multiple categories were Clu, Hsp90aa1, Avp, Dlg4, Il33, Rnf6 (number of participated categories were 8, 7, 6, 6, 6 and 6). Avp mediated apoptosis in various cells (Higashiyama et al., 2001; Wahlstrom et al., 2004), and was regulated by lncRNAs Lhx1os, RP23-442I7.1, and XLOC_085831. Dlg4 (also known as PSD95) was involved in the disruption of synaptic structure and neuronal apoptosis (Bustos et al., 2017; Sultana, Banks & Butterfield, 2010), and had the largest number of interaction pairs in neurological system alternation, metabolism alternation, cellular communication, molecular modification, and behavior changes. All the top five relevant lncRNAs had interacted with Avp and Dlg4, suggesting the crucial roles of related ceRNA networks in PND pathogenesis.

Conclusion

In the present study, we identified DElncRNAs in the hippocampus of aged mice during the development of PND, and established lncRNA-associated ceRNA regulatory networks in eight hippocampal pathological processes including neurological system alteration, metabolism alteration, etc. The results suggest that lncRNAs play roles in PND-related CNS pathologies through the ceRNA network, and provide potential lncRNA-microRNA-microRNA ceRNA network pathways and therapeutic targets for PND research and treatment.

Supplemental Information

Supplemental Information 1 Primer sequences.

Supplemental Information 2 Raw data.

Supplemental Information 3 Raw data for freezing time (Figure 1).

Supplemental Information 4 MIQE checklist.

Supplemental Information 5 Code for volcano plots.

Supplemental Information 6 Code for differential expression analysis.

Supplemental Information 7 Code for GO and KEGG analysis.

Supplemental Information 8 Code for RNAs interaction analysis.

Additional Information and Declarations

Competing Interests

Author Contributions

Animal Ethics

Data Availability

The authors declare that they have no competing interests.

Bowen Zhou performed the experiments, analyzed the data, authored or reviewed drafts of the article, and approved the final draft.

Yuxiang Zheng performed the experiments, analyzed the data, authored or reviewed drafts of the article, prepared figures and/or tables, and approved the final draft.

Zizheng Suo analyzed the data, prepared figures and/or tables, and approved the final draft.

Mingzhu Zhang analyzed the data, prepared figures and/or tables, and approved the final draft.

Wenjie Xu analyzed the data, prepared figures and/or tables, and approved the final draft.

Lijuan Wang analyzed the data, prepared figures and/or tables, and approved the final draft.

Dazhuang Ge analyzed the data, prepared figures and/or tables, and approved the final draft.

Yinyin Qu conceived and designed the experiments, analyzed the data, authored or reviewed drafts of the article, and approved the final draft.

Qiang Wang conceived and designed the experiments, authored or reviewed drafts of the article, and approved the final draft.

Hui Zheng conceived and designed the experiments, authored or reviewed drafts of the article, and approved the final draft.

Cheng Ni conceived and designed the experiments, authored or reviewed drafts of the article, and approved the final draft.

The following information was supplied relating to ethical approvals (i.e., approving body and any reference numbers):

The animal study was reviewed and approved by Biomedical ethics committee of Cancer Hospital, Chinese Academy of Medical Sciences and Peking Union Medical College (NCC2021A040).

The following information was supplied regarding data availability:

The data is available at NCBI Sequence Read Archive: PRJNA1054804.

The code is available in the Supplemental Files.

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
