# Peer review of "The role of lncRNAs related ceRNA regulatory network in multiple hippocampal pathological processes during the development of perioperative neurocognitive disorders"

_PeerJ, doi:10.7717/peerj.17775_

## Round 0.1 · original submission · Major Revisions

In addition to the revisions requested by the reviewers, the authors should also expand the methodology details (non-exhaustive, illustrative examples: What version of R and other packages mentioned were used? For binding energy predictions, what scores, stats, and significance thresholds were used? What version of Perl was used? Which stats were performed with Graphpad, as some were with R and/or Perl? etc). Additionally, code should be made available as discussed in the 'Data and Materials Sharing' Policy (https://peerj.com/about/policies-and-procedures/#data-materials-sharing).

**Language Note:** The review process has identified that the English language must be improved. PeerJ can provide language editing services - please contact us at [email protected] for pricing (be sure to provide your manuscript number and title). Alternatively, you should make your own arrangements to improve the language quality and provide details in your response letter. – PeerJ Staff

Reviewer 1 ·

Basic reporting

Perioperative neurocognitive disorders (PND) pose a significant challenge in elderly surgical patients, with implications for postoperative cognitive function. This study investigates the regulatory role of long non-coding RNAs (lncRNAs) in PND through a comparative transcriptome analysis of the hippocampus in mice. Differential expression analysis revealed distinct patterns of lncRNAs, microRNAs, and mRNAs associated with some potential neuron-related functions such as neurological system alteration, neuronal development, and behavior changes. Furthermore, the construction of lncRNA-microRNA-mRNA competing endogenous RNA (ceRNA) networks provided insights into the regulatory interactions underlying PND pathogenesis. Notably, the ceRNA networks highlighted the potential roles of specific lncRNAs, microRNAs, and mRNAs in modulating key pathways linked to cognitive dysfunction in the aging brain. These findings contribute to our understanding of the molecular mechanisms driving PND and offer potential targets for therapeutic interventions. The manuscript is straightforward and easy to follow. I only have a few comments that I hope might be useful to help the authors further strengthen their manuscript.

Experimental design

no comment

Validity of the findings

no comment

Additional comments

1. Owing to the extensive array of tests conducted for the assessment of differential expression, the reviewer recommends the implementation of measures to control the false discovery rate associated with these events.

2. In Figure 7, the authors investigate the function of DE-mRNAs. The reviewer suggests performing Gene Set Enrichment Analysis (GSEA) and KEGG (Kyoto Encyclopedia of Genes and Genomes) analysis in addition to the Gene Ontology (GO) analysis to compare the results.

3. Following question 2, the reviewer recommends conducting the aforementioned analyses for Figures 4-6. This would assist the authors in providing a more detailed discussion of the specific networks.

4. For all network diagrams, the reviewers suggest incorporating information on the degree for each node and visually representing the size of those nodes. Additionally, defining the hub of the network may offer further insights into the biological networks.

5. There are some minor editorial adjustments that can be easily addressed. For instance, in the method section of the abstract, "qPCR were..." should be corrected to "qPCR was..."; and the labeling, such as fig./Fig., should be made consistent throughout the manuscript.

6. Line 350: The reviewer is puzzled by the phrase "top 216 annotated..."; is this referring to only 216 pairs, or did the authors select the top 216 pairs from a larger set? In the latter case, could the authors provide clarification on the criteria for this selection?

7. The authors conducted experimental validation of DElncRNA. The reviewer recommends presenting the results graphically in a new figure and incorporating it into Figure 10.

·

Basic reporting

Enhancement of Clarity and Readability:
The article would benefit from enhancing the clarity and readability of both the English language used throughout and the figures presented. Ensuring precise language and clear figures will improve the overall accessibility of the research findings.

Experimental design

Explanation of Animal Selection: The authors should provide a clear rationale for exclusively using female animals in their study. Understanding the reasoning behind this decision would provide valuable context for interpreting the experimental results and addressing potential gender-related implications.
Elaboration on Control Group: A more detailed explanation of the control group is necessary. Specifically, the authors should clarify whether the animals received identical anesthesia and, if not, how they differentiate the effects of anesthesia from those of the surgical procedure.
Rationale for Fear Conditioning Test: The authors should justify their choice of the fear conditioning test over other potential tests.
Focus on the Hippocampus: The authors should elaborate on the rationale behind focusing on the hippocampus in their study. A clear explanation of why this brain region was chosen as the primary focus would provide valuable context for interpreting the significance of the findings and their implications for understanding neural mechanisms.
Confirmation Experiments with mRNAs: Including confirmation experiments and validating their results with a set of mRNAs will strengthen the robustness of the study findings.
Scheme of Potential Mechanisms of Action of lncRNA: It would be beneficial for the authors to include a scheme illustrating the potential mechanisms of action of the lncRNA under investigation. Visual representation of these mechanisms would aid in conceptualizing the proposed hypotheses and understanding the underlying biological processes.

Validity of the findings

-

Additional comments

-

---

## Round 0.2 · Major Revisions

Thank you for your revisions.

1. While the reviewer and I assessed that the revisions addressed the major comments, some grammatical issues remain that will still need to be addressed.

Additionally, during the editorial decision, we discovered the following issues which must be fully addressed:

2. This paper appears to be highly similar:
Zhang M, Suo Z, Qu Y, Zheng Y, Xu W, Zhang B, Wang Q, Wu L, Li S, Cheng Y, Xiao T, Zheng H, Ni C. Construction and analysis of circular RNA-associated competing endogenous RNA network in the hippocampus of aged mice for the occurrence of postoperative cognitive dysfunction. Front Aging Neurosci. 2023 Mar 27;15:1098510. doi: 10.3389/fnagi.2023.1098510.
Please explain how the present study differs.

3. These papers maybe should have been cited; please confirm and add if appropriate or provide an explanation if not:
Pant, Tarun PhD*; DiStefano, Johanna K. PhD†; Logan, Sara PhD‡; Bosnjak, Zeljko J. PhD*,§. Emerging Role of Long Noncoding RNAs in Perioperative Neurocognitive Disorders and Anesthetic-Induced Developmental Neurotoxicity. Anesthesia & Analgesia 132(6):p 1614-1625, June 2021. | DOI: 10.1213/ANE.0000000000005317
https://www.frontierspartnerships.org/articles/10.3389/adar.2022.10831/full
https://www.frontiersin.org/articles/10.3389/fncel.2022.1024475/full
https://psycnet.apa.org/record/2023-64191-001

4. There appear to be other papers that should have been cited. The authors need to do a careful literature search for recent articles and resubmit with proper citations.

5. Also, in the abstract conclusion, we suggest rephrasing to:
"In summary, we have constructed the lncRNA-associated ceRNA network during PND development in mice, explored the role of lncRNAs in multiple pathological processes in the mouse hippocampus, and provided insights into the potential mechanisms and therapeutic gene targets for PND."

A careful review of the grammar, citations, etc. must be performed and adequately addressed.

Reviewer 1 ·

Basic reporting

no comment

Experimental design

no comment

Validity of the findings

no comment

Additional comments

The authors have made revisions to their manuscript in accordance with my feedback, and I am content with the revised version, thereby considering it suitable for acceptance. While there are still some grammatical issues, these can be resolved during the finalization of the manuscript.

---

## Round 0.3 · accepted · Accept

Thank you for addressing these additional items. We are pleased to accept your article for publication.